



# 1 A new theoretical interpretation of Archie's saturation exponent

Paul W.J. Glover[1]
[1]School of Earth and Environment, University of Leeds, UK
*Correspondence to*: Paul W.J. Glover (P.W.J.Glover@Leeds.ac.uk)
**Abstract.** This paper describes the extension of the concepts of connectedness and conservation of connectedness that underlie
the generalised Archie`s law for *n* phases to the interpretation of the saturation exponent. It is shown that the saturation
exponent as defined originally by Archie arises naturally from the generalised Archie's law. In the generalised Archie`s law
the saturation exponent of any given phase can be thought of as formally the same as the phase (i.e., cementation) exponent,
but with respect to a reference subset of phases in a larger *n*-phase medium. Furthermore, the connectedness of each of the
phases occupying a reference subset of an *n*-phase medium can be related to the connectedness of the subset itself by
$G_i = G_{ref} S_i^{n_i}$ . This leads naturally to the idea of the term $S_i^{n_i}$ for each phase *i* being a fractional connectedness, where the
fractional connectednesses of any given reference subset sum to unity in the same way that the connectednesses sum to unity
for the whole medium. One of the implications of this theory is that the saturation exponent of any phase can be now be
interpreted as the rate of change of the fractional connectedness with saturation and connectivity within the reference subset.

## 15 1 Introduction

Currently, there is no well-accepted physical interpretation of the saturation exponent other than qualitatively as some measure
of the efficiency with which electrical flow takes place within the water occupying a partially saturated rock. Some might say
that the meaning is not important as long as one can reliably obtain the water saturation of reservoir rocks with sufficient
accuracy to calculate reserves. According to the 2016 BP Statistical Review of World Energy (BP, 2016), the world had proved
oil reserves at the end of 2015 of 1.6976 trillion (million million; Tbbl.) barrels, slightly down on the value at the end of 2014
(1.7 Tbbl.) and significantly above the respective values at the end of 1995 (1.1262 Tbbl.) and 2005 (1.3744 Tbbl.). The same
source lists proven natural gas reserves of 186.9 trillion cubic metres (Tcm) at the end of 2015, slightly lower than at the end
of 2014 (187.0 Tcm) and significantly and progressively higher than the values at the end of 1995 (119.9 Tcm) and 2005
(157.3 Tcm). This represents combined oil and gas reserves of approximately 78.4 trillion US dollars combined at end
December 2015 prices (using WTI crude and Henry Hub).
Even a tiny uncertainty of, say, 0.01 in a saturation exponent of 2 (i.e., 0.5% or 2±0.01) would result in an error in the
reserves of about ±254.36 billion US dollars; the equivalent of 82 Queen Elizabeth class aircraft carriers or one mission to
Mars. This calculation has been carried out by calculating the percentage change in hydrocarbon saturation resulting from an
error of 2±0.01 in the value of the saturation exponent. Since the calculated change in hydrocarbon saturation also depends on
other parameters in Archie's equations. Typical representative values for these parameters have had to be used, and these are
$R_T = 500\ \Omega.m, R_w = 1\ \Omega.m, \phi = 0.1$ and $m = 2$. When these values are used with $n = 2 \pm 0.01$ a change of ±0.3245% was calculated
for the hydrocarbon saturation, allowing the change in global reserves to be calculated.
Within the hydrocarbon industry it is extremely common to assume that the saturation exponent is about 2 for most rocks.
However, it is worthwhile thinking about the 254 billion dollar global shortfall in revenue if it really is equal to 2.01 instead.
These frightening large financial values make it extremely important that the physical interpretation of the saturation exponent
in the classical Archie's law is well understood. This paper attempts to provide a new theoretical and physical interpretation.
The classical Archie`s laws (Archie, 1942) link the electrical resistivity of a rock to its porosity, to the resistivity of the water
saturating its pores, and to the fractional saturation of the pore space with the water. They have been used for many years to



calculate the hydrocarbon saturation of the reservoir rock and hence hydrocarbon reserves. The classical Archie's laws contain
two exponents, *m* and *n*, which Archie called the cementation exponent and the saturation exponent, respectively. The
conductivity of the hydrocarbon saturated rock is highly sensitive to changes in either exponent.
Like the cementation exponent, and despite its importance to reserves calculations, the physical meaning of the saturation
exponent is difficult to understand from a physical point of view, which leads to petrophysicists not giving it the respect it
deserves. It is common, for example, to hear that, in the absence of laboratory measurements, the saturation exponent has been
taken to be equal to 2, which it has just been noted is bound to lead to gross errors. While it is true that there seems to be a
strong preference for values of saturation exponent near $2\pm0.5$ for most water-wet rocks, oil-wet rocks show much higher
values (4-5) (Montaron, 2009; Sweeney and Jennings, 1960), and there is evidence that the saturation exponent changes with
saturation, with the type of rock microstructure and with saturation history, leading to hysteresis in the plot of resistivity index
as a function of water saturation.
When a saturation exponent is derived from laboratory measurements, it is commonly done by fitting a straight line to
resistivity data where the y-axis is the logarithm of the measured rock resistivity divided by the resistivity of the saturating
water and the x-axis is the logarithm of the water saturation. The problem is that the saturation exponent varies with water
saturation, becoming significantly smaller at low saturations, leading to an uncertainty in which value to use. This observation
also gives us the first hint that it is the connectedness of the water phase that is controlling the saturation exponent just as it
did for the phase exponent in the generalised Archie's law.
It is clear that the physical understanding of the saturation exponent needs to be improved. The purpose of this paper is to
investigate the elusive physical meaning of the saturation exponent, where it is shown that the saturation exponents are
intimately linked to the phase exponents in the generalised Archie's model.

## 2 Traditional interpretations

Considering the classical form of Archie's laws; the first Archie law relates the formation factor *F*, which is the ratio of the
resistivity of a fully saturated rock $\rho_o$ ($R_o$) to the resistivity of the fluid occupying its pores $\rho_f$ ($R_w$), to the rock porosity $\phi$ and
a parameter he called the cementation exponent *m*, where the symbols in parentheses are those traditionally used in the
hydrocarbon industry. Archie's first law can be expressed as $F = \rho_o/\rho_f = \phi^{-m}$ using resistivities (Archie, 1942), or as
$G = \sigma_o/\sigma_f = \phi^{+m}$ using conductivities. In the latter case *G* is called the conductivity formation factor or the connectedness
(Glover, 2009). It can easily be seen that the effective resistivity and effective conductivity of the fully saturated rock can be
expressed as $\rho_o = \rho_f \phi^{-m}$ and $\sigma_o = \sigma_f \phi^{+m}$ using resistivities or conductivities, respectively. It should be noted that this work
does not consider the form of Archie's law which includes the so-called 'tortuosity factor' *a*, which was developed by Winsauer
et al. (1952). The role of this parameter is discussed fully in Glover (2016).
Archie's second law considers that the rock is not fully saturated with a conductive fluid, but is partially saturated with a
fractional water saturation $S_w$. It relates the resistivity index *I*, which is the ratio of the resistivity of a partially saturated rock
$\rho_{eff}$ to the resistivity of the fully saturated rock $\rho_o$, to the water saturation $S_w$ and a parameter he called the saturation exponent
*n*. Archie's second law can be expressed as $I = \rho_{eff}/\rho_o = S_w^{-n}$ using resistivities, or $1/I = \sigma_{eff}/\sigma_o = S_w^{+n}$ using
conductivities.
The two laws may be combined to give $\rho_{eff} = \rho_f \phi^{-m} S_w^{-n}$ using resistivities, and $\sigma_{eff} = \sigma_f \phi^{+m} S_w^{+n}$ if conductivities are
used. In reserves calculations, the resistivity of the partially saturated rock, the resistivity of the pore water, the porosity of the
rock and the two exponents are "known" from logging or laboratory measurements. This enables the water saturation $S_w$ and
hence the hydrocarbon saturation $S_h = (1 - S_w)$ and, consequently, the reserves to be calculated.





Archie's laws require that both the rock matrix and all but one of the fluid phases that occupy the pores to have infinite
resistivity. Hence, it is a model for the distribution of one conducting phase (the pore water) within a rock sample consisting
of a non-conducting matrix and other fluids which also have zero or negligible conductivity. Problems arise when there are
other conducting phases in the rock, such as clay minerals. These problems have generated a huge amount of research in the
past (e.g., Waxman and Smits, 1968; Clavier *et al.*, 1984), which are reviewed in Glover (2015). The classical Archie`s laws
were based upon experimental determinations, however, progressive theoretical work (Sen *et al.*, 1981; Mendelson and Cohen,
1982; Glover et al., 2000a; Glover, 2009) arising from the need to have versions of Archie's law which were valid for a
conductive matrix (Glover et al., 2000b) has underpinned these initially empirical relationships, culminating in a generalised
Archie`s law which was published in 2010 (Glover, 2010).
**3 The generalised Archie's law**
The generalised Archie's law (Glover, 2010) extends the classical Archie's law to a porous medium containing *n* phases. It is
based on the same concept of connectedness that was introduced in the author's previous interpretation of the cementation
exponent (Glover, 2009). In the 2009 paper the connectedness was defined as
$$G \equiv \frac{\sigma_o}{\sigma_w} = \frac{1}{F} = \phi^m \ , \tag{1}$$

where $F$ is the formation factor. The connectedness of a given phase is a physical measure of the availability of pathways for
conduction through that phase. The connectedness is the ratio of the measured conductivity to the maximum conductivity
possible with that phase (i.e., when that phase occupies the whole sample). This implies that the connectedness of a sample
composed of a single phase is unity. Connectedness is not the same as connectivity. The connectivity is defined as the measure
of how the pore space is arranged in its most general sense as that distribution in space which makes the contribution of the
specific conductivity of the material express itself as a different conductance (see Glover, 2010). The connectivity is given by
$\chi = \phi^{m-1}$ and depends upon the porosity and the classical Archie's cementation exponent *m*. It should be noted that the
connectedness is also given by
$$G = \phi\chi \ , \tag{2}$$

and then it becomes clear that the connectedness depends both upon the amount of pore space (given by the porosity) and the
arrangement of that pore space (given by the connectivity).
The generalized Archie's law was derived by Glover (2010) and is given by
$$\sigma = \sum_i \sigma_i \phi_i^{m_i} \qquad \text{with} \qquad \sum_{i=1} \phi_i = 1 \ , \tag{3}$$

where there are *n* phases, each with a conductivity $\sigma_i$, a phase volume fraction $\phi_i$ and an exponent $m_i$. The porosity and
cementation exponent in the classical Archie's law are the same as the pore space phase volume fraction and pore space phase
exponent in the generalized Archie's law, respectively. However, the pore space and the matrix may be subdivided into any
number of other phases as required. Indeed, the generalized Archie's law will not contain a term that represents the pore space
unless the pore space is only occupied by a single phase.
In the generalized law the phase exponents can take any value from zero to infinity. Values less than unity represent a
phase with an extremely high degree of connectedness, such as that for the solid matrix of a rock. Connectedness decreases as
the phase exponent increases. Phase exponents that tend towards 1 are associated with a highly connected phase which is
analogous to the low cementation exponents occurring in the traditional Archie's law for networks of high aspect ratio cracks.
Phase exponents about 2 represent the degree of connectedness that one might find when the phase is partially connected in a
similar way to which the pore network in a sandstone is connected, and which is, again, analogous to that scenario in the





traditional Archie's law. By extension, higher values of phase exponents represent lower phase connectedness, such as that in
the traditional Archie's law for the pores in a vuggy limestone.

It is clear that the classical and generalized laws share the property that the exponents modify the volume fraction of the

relevant phase with respect to the total volume of the rock. However the exponents in the generalized law differ from the
classical exponent because some of them have values which are not measureable because their phases are composed of
materials with negligible conductivity. Despite this, each phase has a well-defined exponent providing (i) it has a non-zero
volume fraction and (ii) the other phases are well-defined.

It should be noted that higher phase exponents tend to be related to lower phase fractions, although this relationship is not

implicit in the generalized Archie's law as it is currently formulated.

It is important to consider Equation 1 and Equation 4 together to develop a fuller understanding of the model. There is an

infinite number of solutions to Equation 4 even in the most restrictive 2 phase system. However, there is only a small subset
of solutions if both Equation 1 and Equation 4 are to be fulfilled together, as the model requires. The problem of having enough
degrees of freedom is not problematic for 3 phases or more, and is trivial for one phase. Consequently, if there is to be a
problem with the Glover (2010) model, it should be clearest for a two phase system.

Considering a two phase system. Equation 1 gives $\phi_1 = 1 - \phi_2$ while Equation 4 can be written as $\phi_1^{m_1} + \phi_2^{m_2} = 1$.

Substituting we obtain either $(1 - \phi_2)^{m_1} + \phi_2^{m_2} = 1$ or $(1 - \phi_1)^{m_2} + \phi_1^{m_1} = 1$. These equations are formally the same.
They each have trivial solutions when each of the volume fractions tends to unity, the other volume fraction consequently
tending to zero. Another solution occurs when $m_1 = m_2 = 1$, which is the simple parallel conduction model. Only one other
solution exists for the general case where the volume fractions are variable, and that requires $m_1 > 1$ when $m_2 < 1$ or vice versa.
Consequently the non-trivial solution for a 2-phase medium falls into one of the following classes:
(i)        $m_1 = m_2 = 1$, the phases, whatever their volume fractions, are arranged in parallel and both have a unity exponent.
(ii)       $m_1 > 1$ and $m_2 < 1$. This implies that Phase 1 has a path across the 3D medium that is less connected than a parallel

arrangement of that phase. Since we have a two phase medium Phase 2 must have a path across the medium which

is more connected than a parallel arrangement, hence forcing $m_2 < 1$.

(iii)      $m_1 < 1$ and $m_2 > 1$. Since the system is symmetric. This scenario is formally the same as (ii) above, but with the

phase numbers switched around.

Consequently, for a two-phase medium, defining the porosity and connectedness (or exponent) of one of the phases
immediately fully defines the other phase. For higher numbers of phases, there are more solutions, but if the porosity and
connectedness (or exponent) of $n$-1 of the phases is known, the nth phase is also fully defined in the same way. The logical
extension of this idea is that both the sum of the volume fractions of the $n$-phases is unity and the sum of the connectednesses
of the $n$-phases is also unity, or that both volume fraction and connectedness are conserved in a three-dimensional $n$-phase
mixture. Another, more intuitive way of looking at this is as follows. It has already been shown that the connectedness of a
system that contains only one phase is unity as a result of Equation 1, i.e., if there is one phase $\phi = 1$ and hence $G = 1$. Let us
imagine that a second phase is introduced. Intuitively, it seems reasonable that as the phase fraction of the new phase increases,
its connectedness will increase, and that when this happens both the volume fraction and connectedness of the first phase will
decrease. The same would be true if any number of new phases were introduced – all the phases would compete for a fixed
amount of connectedness, its increase for one phase being balanced by a decrease in at least one of the other phases. In other
words there is a fixed maximum amount of connectedness possible in a three-dimensional sample, expressed by Glover (2010)
as
$$\sum_i \phi_i^{m_i} = \sum_i G_i = 1 .$$
(4)





In summary, both the sum of the volume fractions and the sum of the connectednesses of the phases composing a 3D
medium is equal to unity. The corollary is that connectedness is conserved; if the connectedness of one phase diminishes, there
must be an increase in the connectedness of one or more of the other phases to balance it.
**4 Origin of the saturation exponent**
Within the framework of the classical Archie's laws it is possible to envisage the cementation exponent as controlling how the
porosity is connected within the rock sample volume, and to envisage the saturation exponent as controlling how the water is
connected within that porosity. The cementation exponent is defined relative to the total volume of the rock, while the
saturation exponent is defined relative to the pore space, which is a subset of the whole rock. This is an important concept for
what follows.
The water is one of two phases within the porosity, while that porosity is one of two phases within the rock. Hence, there
exists a three phase system to which the generalised Archie's law can be applied. In fact, the generalised Archie's law can be
used to show that the saturation exponents arise naturally and have a physical meaning: they are defined in the same way as
the phase exponents but are expressed relative to the pore space instead of the whole rock.
By writing the generalized law (Equation 4) for three defined phases; let's say matrix, water and hydrocarbon gas, and
assuming that neither the matrix nor the gas is conductive, i.e., $\sigma_m = 0$ and $\sigma_h = 0$, but allowing the pore space to be partially
saturated with water such that $\phi_h \neq 0$, it is possible to obtain $\sigma_{eff} = \sigma_f \phi_f^{m_f}$. Since $\phi_h \neq 0$, the pore space is partially saturated
with hydrocarbon and partially saturated with water. It is also possible to write $\phi_f = \phi S_w$, and hence obtain
$$\sigma_{eff} = \sigma_f \phi^{m_f} S_w^{m_f} \ . \tag{5}$$

Comparison with the classical Archie's laws, which can be written as $\sigma_{eff} = \sigma_f \phi^m S_w^n$ (Tiab and Donaldson, 2004) shows
structural similarity. However, the exponent $m_f$ in Equation 5 is expressed relative to the whole rock because it is the phase
exponent for the fluid that appears in Equation 4. By contrast, the cementation exponent $m$ in the classical second Archie's
law is expressed relative to the whole rock because it is defined in Archie's first law, the saturation exponent $n$ is expressed
relative to a subset of the whole rock called the pore space. This idea can be tested easily by imagining whether the saturation
exponent is independent of any changes one might make to the rock matrix: It is possible to see that the saturation exponent is
independent of the rock matrix, and is only sensitive to changes occurring within the pore space.
Both equations provide a valid measure of the effective rock conductivity, so they may be equated, hence obtaining
$\phi^m = \phi^{m_f} S_w^{(m_f - n_f)}$, which can be rearranged to give $G_f = G_{pore} S_w^{n_f}$. Here it can be recognized that the classical Archie
saturation exponent refers to saturation with water and is hence renamed as $n_f$. Since the system is symmetric this equation for
the gas phase (and any other phase that may be present) can be written as
$$G_i = G_{ref} S_i^{n_i} \ , \tag{6}$$

with the pore connectedness being relabeled as the reference connectedness because the equation is valid not only for multiple
phases that fill the porosity, but multiple phases composing any other phase.
Equation 6 gives the connectedness of the $i^{th}$ phase in an $n$-phase 3D medium as depending on both its fractional saturation
$S_i$ within a larger volume which has a connectedness $G_{ref}$ and that reference connectedness. The distribution of that saturation
is taken into account by the exponent $n_i$, which will have a general functional form.





If one considers the whole 3D $n$-phase medium (i.e., one where $\sum_i \phi_i = 1$), Equation 1 states that the connectedness of
each phase is the volume fraction of that phase raised to the value of its phase exponent, and Equation 4 states that the sum of
the connectednesses is unity.
If a subset of a whole $n$-phase medium (i.e., one where $\sum_i \phi_i < 1$) is considered, and labelled the reference subset, the
reference subset will have a connectedness $G_{ref} = \phi_{ref}^{m_{ref}}$ relative to the whole rock, and the connectedness of any phase which
partially occupies the reference subset (e.g., water within the pore space, clay within the rock matrix etc.) is equal to the
connectedness of the reference phase multiplied by the volume fraction of the phase within the reference subset (i.e., the
saturation relative to the reference subset) raised to the value of its saturation exponent.
The definition above is somewhat complex due to the requirement to be both completely general and precise, and that there
are two reference frames here. The first is the whole 3D $n$-phase medium. The second is the 3D reference subset which may
contain between 2 and $n$-1 phases. Conversion between the two reference frames can be carried out using the relationship
$$\phi_i^{m_i - n_i} = \phi_{ref}^{m_{ref} - n_i}, \tag{7}$$

It can also be shown that (Glover, 2010)
$$\sum_i S_i^{n_i} = 1, \tag{8}$$

where the sum is carried out over all the phases within the reference subset.
It should be noted that Equation 8 is formally the same as Equation 4 except that Equation 7 is valid for the reference subset
of phases, while Equation 4 is valid for the whole $n$-phase medium. Hence it is possible to use $S_i = \phi_i / \phi_{ref}$ to write both
Equation 4 and Equation 8 as
$$\sum_i \left( \frac{\phi_i}{\phi_{ref}} \right)^{m_i} = 1. \tag{9}$$

• For a whole $n$-phase medium $\phi_{ref} = 1$ and Equation 9 becomes equal to Equation 4.
• For a subset of the $n$-phase medium $\phi_{ref} < 1$ and Equation 9 becomes equal to Equation 8.
The distinction between the phase exponent and saturation exponent becomes trivial; they each control how connected the
phase is relative to the reference volume fraction. In other words, the transformation $1 \leftrightarrow \phi_{ref}$ leads to $\phi_i \leftrightarrow S_i$ and
$m_i \leftrightarrow n_i$.
There is of course the possibility that the whole $n$-phase medium is itself a subset of a larger medium with more phases. In
this case Equation 9 still holds, but with $\phi_{ref} > 1$.
Hence, both the phase (cementation) exponent and the saturation exponent control how the phase is connected. The phase
exponent does this with reference to the whole rock, while the saturation exponent does it with reference to a subset of the
whole rock. The underlying physical meaning of the saturation exponent is the same as that of the phase (cementation)
exponent, it is only the reference frame that changes. The implication is that the general Archie's law replaces both of the
classical Archie's laws. For an application to a sandstone gas reservoir, one would use a 3 phase generalized Archie law.
Equation 6 is easily transformed to provide a calculable value for the saturation exponent
$$n_i = \frac{\log(G_i) - \log(G_{ref})}{\log(S_i)} = \frac{m_i \log(\phi_i) - m_{ref} \log(\phi_{ref})}{\log(\phi_i) - \log(\phi_{ref})}. \tag{10}$$



## 5 Fractional connectedness


The connectedness $G$ is the inverse of the Archie's formation factor and is central to the generalized Archie's law. The inverse
of the Archie's resistivity (saturation) index $1/I = S_w^n$ is also rather important. It relates the connectednesses of each phase
with respect to the whole rock to the connectedness of the reference subset in Equation 6, and when summed over all the
phases that occupy the reference subset it produces unity as in Equation 8. In this paper the inverse of the Archie's resistivity
(saturation) index has been given the symbol $H_i$ and defined as
$$H_i \equiv S_i^{n_i}. \tag{11}$$

Just as the saturation of any given phase $S_i$ is the ratio of the volume fraction of the phase to that of the all the phases making
up any reference set of phases, $H_i$ is the ratio of the connectedness of the phase to that of the all the phases making up any
reference set of phases. The parameter $H_i$ is in fact a fractional connectedness.
Following the approach of Glover (2009) in the analysis of the cementation exponent, and accepting the deep symmetry
between phase fractions and saturations and between phase exponents and saturation exponents, it is then possible to write
$$n_i = \frac{d}{d\psi}\left(\frac{dH_i}{dS}\right) \tag{12}$$

where $\psi$ is the connectivity of the phase with respect to the reference subset of phases, where
$$\psi = S_i^{n_i - 1}, \tag{13}$$

In this case the rate of change of fractional connectedness with saturation is the product of the saturation exponent and the
connectivity with respect to the reference subset
$$\frac{dH_i}{dS_i} = n_i \psi_i, \tag{14}$$

and the fractional connectedness is the product of the saturation and the connectivity with respect to the reference subset
$H_i = S_i \psi_i$.
Hence, the saturation exponents obey the same laws as the phase (cementation) exponents, but whereas the phase exponents
are defined relative to the whole rock, the saturation exponents are defined relative to some subset of the rock. Table 1 shows
the relationships of the generalised Archie's law expressed relative to the whole rock and with respect to a reference subset of
the whole rock.
For petrophysicists the reference subset has been the porosity and there has only been one conducting phase that partially
saturates that porosity – the pore water. Now we are not restricted to that model. The reference subset could be, for example,
the solid matrix in which a number of separate mineral phases can be defined, one of which might be, say, a target ore.

## 6 Conclusions


The main conceptual steps in this paper are summarised as:
• The classical Archie saturation exponent arises naturally from the generalised Archie's law.
• The saturation exponent of any given phase can be thought of as formally the same as the phase (i.e., cementation)
exponent, but with respect to a reference subset of phases in a larger $n$-phase medium.
• The connectedness of each of the phases occupying a reference subset of an $n$-phase medium can be related to the
connectedness of the subset itself by $G_i = G_{ref} S_i^{n_i}$.



• The sum of the connectednesses of a 3D $n$-phase medium is given by $\sum_i \phi_i^{m_i} = 1$ , mirroring the relationship for phase
volumes $\sum_i \phi_i = 1$ .
• Connectedness is conserved in a 3D $n$-phase medium. If one phase increases in connectedness the connectedness of one
or more of the other phases must decrease to compensate for it, just as phase volumes are conserved with the decrease in
one leading to the increase of another phase.
• The sum of the fractional connectednesses (saturations) of an $n$-phase medium is given by $\sum_i S_i^{n_i} = 1$ .
• Fractional connectedness is conserved in a 3D $n$-phase medium.
• The saturation exponent may be calculated using the relationship $n_i = \dfrac{m_i \log(\phi_i) - m_{ref} \log(\phi_{ref})}{\log(\phi_i) - \log(\phi_{ref})}$ .
• The connectivity of any phase with respect to the reference subset is given by $\psi_i = S_i^{n_i - 1}$ .
• The connectedness of a phase with respect to a reference subset (also called the fractional connectedness) is given by
$H_i = S_i \psi_i$ and depends upon the fractional volume of the phase divided by that of the reference subset (i.e., its saturation)
and the arrangement of the phase within the reference subset (i.e., its connectivity with respect to the reference subset).
• The rate of change of fractional connectedness with saturation $\dfrac{dH_i}{dS_i} = n_i \psi_i$ depends upon the connectivity with respect
to the reference subset $\psi_i$ and the saturation exponent $n_i$.
• Hence, the saturation exponent is interpreted as being the rate of change of the fractional connectedness with saturation
and connectivity within the reference subset, $n_i = \dfrac{d^2 H_i}{d\psi_i dS_i}$ .





**Table 1.** Comparison of all the parameters in the classical and generalised Archie's laws.

| Parameter | Generalised Archie's law | | Classical Archie's law | |
|---|---|---|---|---|
| | **With respect to the whole medium** | **With respect to a reference subset of the whole medium** | **First law** | **Second Law** |
| Phase volume fraction | $\phi_i$ $$\phi_i = \phi_{ref}\, S_i$$ | $S_i$ $$S_i = \phi_i/\phi_{ref}$$ | $\phi$ $$V_f = V_{pore}\, S_w$$ | $S$ $$S_w = V_f/V_{pore}$$ |
| Exponent | $$m_i = \frac{d}{d\chi}\left(\frac{dG_i}{d\phi}\right)$$ $$m_i = \frac{\log(\sigma_i) - \log(\sigma_f)}{\log(\phi_i)}$$ | $$n_i = \frac{d}{d\psi}\left(\frac{dH_i}{dS}\right)$$ $$n_i = \frac{m_i\log(\phi_i) - m_{ref}\log(\phi_{ref})}{\log(\phi_i) - \log(\phi_{ref})}$$ | $m$ $$m = \frac{\log(\sigma_{eff}) - \log(\sigma_f)}{\log(\phi)}$$ | $n$ $$n = \frac{\log(\sigma_{eff}) - \log(\sigma_{100})}{\log(S_w)}$$ |
| Connected-ness | $$G_i \equiv \phi_i^{m_i}$$ $$G_i = \phi_i \chi_i$$ $$G_i = 1/F_i$$ $$G_i = G_{ref}\, H_i$$ | $$H_i \equiv S_i^{n_i}$$ $$H_i = S_i \psi_i$$ $$H_i = 1/I_i$$ $$H_i = G_i/G_{ref}$$ | undefined | undefined |
| Connect-ivity | $$\chi = \phi_i^{m_i - 1}$$ | $$\psi = S_i^{n_i - 1}$$ | $$\chi = \phi^{m-1}$$ | undefined |
| Rate of change of connected-ness | $$\frac{dG_i}{d\phi_i} = m_i \chi_i$$ | $$\frac{dH_i}{dS_i} = n_i \psi_i$$ | undefined | undefined |
| Sum of phases | $$\sum_{i=1}\phi_i = 1$$ $$\sum_i S_i > 1$$ | $$\sum_{i=1}\phi_i < 1$$ $$\sum_i S_i = 1$$ | $$\phi_{pore} + \phi_{matrix} = 1$$ | $$S_w + S_o + S_g = 1$$ |
| Sum of connected-nesses | $$\sum_i \phi_i^{m_i} = \sum_i G_i = 1$$ $$\sum_i\left(\frac{\phi_i}{\phi_{ref}}\right)^{m_i} = 1$$ The transformation $1 \leftrightarrow \phi_{ref}$ leads to $\phi_i \leftrightarrow S_i$ and $m_i \leftrightarrow n_i$ | $$\sum_i S_i^{n_i} = \sum_i H_i = 1$$ | undefined | undefined |
| Effective conduct-ivity | $$\sigma_{eff} = \sum_i \sigma_i \phi_i^{m_i}$$ | $$\sigma_{eff} = \sum_i \sigma_i \phi_{ref}^{m_i} S_i^{m_i}$$ | $$\sigma_{eff} = \sigma_f \phi^m$$ | $$\sigma_{eff} = \sigma_f \phi^m S_w^n$$ |






**Data availability**

This work is entirely theoretical and contains no data.

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
