# Peer review of "A new theoretical interpretation of Archie's saturation exponent"

_Solid Earth, 2017_

## Referee Comment (RC1) · H. Milsch (Referee) · 13 Mar 2017

**Review on paper manuscript se-2017-5:**

**Summary:**

In this paper and based on earlier findings (Glover, 2009; 2010), the author derives a new theoretical interpretation of the saturation index contained in Archie's second law. The essence of this interpretation is the extension of the "generalized Archie's law" outlined by the author in Glover (2010), where the saturation index is viewed as being "formally the same as the phase exponent, but with respect to a reference subset of phases in a larger n-phase medium".
The author carried out an important task with implications for fundamental rock physics and industrial applications alike. The paper is well structured and, in my perception, mathematically sound and may definitely be suitable for publication in *Solid Earth (SE)*. However, there are a number of substantial issues outlined in the following that I encourage the author to address before the paper can be recommended for publication.

**General comments:**

1. It should be noted that (1) what is attempted here is a *physical* interpretation of an *empirical* parameter, which I find *per se* problematic. Also, it should be noted that (2) the outlined interpretation comes as an *ad hoc* approach and that (3) no proof is presented that this approach and the resulting interpretation is physically correct. Please comment and clarify within the manuscript.

2. The motivation for performing this particular theoretical investigation is well presented in Section 1. However, this discussion also implicitly suggests that reserves calculations can now be performed with unprecedented precision. There is no proof that this is the case. It should also be noted that there will still be experiments and/or analyses to be performed to parameterize the newly introduced equations. How these experiments/analyses should look like and what type of data is required should be included in the text.

3. The theoretical approach is, mathematically, not very demanding but it appears abstract and hard to grasp. I therefore would wish to see (1) some of the equations to be developed in more detail (e.g. in some appendix), (2) one or more graphical representations of the model to better depict the theory, and, not least, (3) a few example calculations where for some type of rock with some kind(s) of fluid(s) some saturation index is derived and then is compared to existing (experimental) data. Please see also comments below.

**Specific comments:**

- Section 3; Lines 148-152: I wonder if this is correct. What about percolation or a percolation threshold? Please comment. This comment also applies to Line 260.

- Section 3: In this section a first illustrating sketch should be introduced.

- Section 4; Line 171: Please clarify from where this equation arises.

- Section 4; Lines 178-180: Reasoning unclear. Please improve.

- Section 4; Lines 182-185 and 202: The equations contained here should be fully derived, e.g. within the section or some appendix.

- Section 4; Lines 199 and following: Here, a second illustrating sketch should be introduced.

- Section 4; Lines 213-214: Can this transformation be exemplified or illustrated?

- Section 4; Eq. (10): This equation should be fully derived and also (numerically) exemplified for a 3-phase medium like the one mentioned before in Line 221.

- Section 5: The motivation for this section is somewhat unclear and should be outlined.

- Section 5; Line 234: Please briefly recall the approach of Glover (2009).

- Section 5; Eq. (12), (13), and (14): In my opinion the derivation should be improved/expanded and also inverted such that Eq. (12) is the final outcome (as in Section 6).

- Section 5; Eq. (12): This equation is only correct if one can assume that $n_i \neq f(\Psi_i)$. Please show that this is the case.

- Section 5; Eq. (12): Please show that Eq. (12) yields Eq. (10) or vice versa.

- Section 5; Lines 248-250: To illustrate this statement and by applying either equation I would wish to see an example calculation / numerical evaluation for a 4-phase porous medium (e.g. quarz, clay, water, gas).

**Technical corrections:**

- the expression "rate of change" suggests some time dependence/derivative and should be replaced throughout the manuscript including in Table 1 by some other, more appropriate, expression

- Lines 51-52: please check if statement is correct

- Glover (2016) not in reference list

- the use of "Φ" (phi) for both porosity and phase volume fractions may lead to confusion. Please reconsider

- Line 125: Equation 4 (?), please check. If correct move Eq. (4) in Line 115 up in text

- Line 131: please check indices in equation

- Line 191: Equation 1 (?), please check

- Line 206: Equation 7 (?), please check

- Lines 237-238: Index "i" missing in "$\Psi$" (psi).

**References:**

Glover, P. W. J.: What is the cementation exponent? A new interpretation, The Leading Edge, 82–85, doi: 10.1190/1.3064150, 2009.

Glover, P. W. J.: A generalised Archie's law for n phases, Geophysics, 75 (6), E247-E265, doi: 10.1190/1.3509781, 2010.

---

## Referee Comment (RC2) · Anonymous Referee #2 · 1 Apr 2017

Dear Editor The comments for se-2017-5 are given as follows:

Comments to the Author Manuscript ID: se-2017-5 Title: A new theoretical interpretation of Archie's saturation exponent Author: Paul W. J. Glover

The classical Archie's law is an important expression to describe the relationship between electrical property and the porosity of rocks. In this paper, the author builds a new theory to extend Archie's law and make it more completely, so that it can be applied to the n-phase medium. Although there is a Table listed to make a comparison of these two theories, I do not think it is enough to show the advances and validities of the new theory. It is better to give a simulated analysis in the paper at least. However, the author has presented that there is no data here, so hope to see the related paper soon, which interests me the best.

[Figure]

Best wishes.

Please also note the supplement to this comment:
http://www.solid-earth-discuss.net/se-2017-5/se-2017-5-RC2-supplement.pdf
─────────────────────────────

---

## Referee Comment (RC3) · G. Heinson (Referee) · 10 Apr 2017

An interesting and worthwhile paper on the importance and calculation of Archie's Law saturation exponent. I have little background in this area of pertrophysics other than accepting the well-known and simple empirical relationship between resistivity, pore fluid, porosity and saturation. Glover explains both the mathematics in a careful manner and also the context for developing such a theoretical approach.

The argument about estimated reserves is both dramatic and perhaps a bit ambit, but it does provide a good reason why a redefinition might matter. Of course, any thing connected with such large reserves and value will have a significant effect as a small percentage.

Line 29 starting "Since..." seems to be missing part of the sentence.

[Figure]

The sentence from Line 82 - 86 is quite long and could be re-phrased.

The flow of logic is reasonably well presented. I'm a bit confused on Line 125 that Equation 1 and 4 are mentioned, but Equation 4 does not get defined until line 155. The example for a two-phase system from Line 130 is good in highlighting a simple case.

The sentence on Line 176 "By contrast..." could be rephrased. My take is that the exponent is related to the fractional volume of pores filled with the fluid rather than being a related to the whole rock. There is a bit of confusing sentence structure.

The conclusions are a nice summary of the paper, but need the paper to make sense of the equations. Thus, they could not really be read stand-alone. Not sure if this is a problem.

---

## Author Comment (AC1) · 9 May 2017

**A new theoretical interpretation of Archie's saturation exponent**

Paul W.J. Glover1

1School of Earth and Environment, University of Leeds, UK

Correspondence to: Paul W.J. Glover (P.W.J.Glover@Leeds.ac.uk)

[revised manuscript text omitted]

Archie's laws require that both the rock matrix and all but one of the fluid phases that occupy the pores to have infinite 83 resistivity. Hence, it is a model for the distribution of one conducting phase (the pore water) within a rock sample consisting 84 of a non-conducting matrix and other fluids which also have zero or negligible conductivity. Problems arise when there are 85 other conducting phases in the rock, such as clay minerals. These problems have generated a huge amount of research in the past (e.g., Waxman and Smits, 1968; Clavier et al., 1984), which are reviewed in Glover (2015). The classical Archie's laws 86 87 were based upon experimental determinations. However, there has been progressive theoretical work (Sen et al., 1981; 88 Mendelson and Cohen, 1982) showing that for at least some values of cementation exponent Archie's law has a theoretical 89 pedigree while hinting that the law may be truly theoretical for all physical values of cementation exponent. A study has 90 recently shown that the Winsauer et al. (1952) modification to Archie's law is only needed to compensate for systematic errors 91 in the measurement of its input parameters and has no theoretical basis (Glover, 2016). Meanwhile independent modifications 92 to the original Archie's law have allowed it to be used when both the pore fill and the matrix have significant electrical 93 conductivities (Glover et al., 2000a; Glover, 2009), such as the case when a rock melt occupies spaces between a solid matrix 94 in the lower crust (Glover et al., 2000b). This has culminated in a generalised Archie's law which is valid for any number of conductive phases in the three-dimensional medium which was published in 2010 (Glover, 2010). 95

**96 3 The generalised Archie's law**

The generalised Archie's law (Glover, 2010) extends the classical Archie's law to a porous medium containing *n* phases. It is based on the same concept of connectedness that was introduced in the author's previous interpretation of the cementation exponent (Glover, 2009). It should be noted that from this point in this paper that the symbol  $\phi$  refers not just to the porosity of the rock, but to the volume fraction of a particular phase, whether it be the matrix, the water, hydrocarbon or whatever other phase may be present. It will either be used for a specific phase such as water (e.g.,  $\phi_i$ ) or for a set of phases (e.g.,  $\phi_i$ ). The unsubscripted symbol continues to refer to conventional porosity, where  $\phi = \sum_i \phi_i - \phi_m$ , and  $\phi_m$  is the phase fraction of the rock matrix (conventionally equal to  $1-\phi$ ). Occasionally, the unsubscripted symbol will also be used when the general properties 104 of phase fractions are being discussed, such as in the following two equations.

[revised manuscript text omitted]

- 174 Figure 1 is an illustrative example of the idea of a fixed amount of connectedness, using a 2D slice for simplicity and 175 clarity. Hence, Figure 1 shows a two dimensional slice through a 3D 4-phase water-wet medium composed of detrital quartz 176 grains, a string of clay, and a porosity that is partially filled with water, at near irreducible saturation and oil. The figure should 177 be read in two columns. The left hand column shows an arbitrary arrangement of the four phases that together completely 178 make up the medium (Fig. 1a). In this case I have chosen to represent the detrital quartz as sub-angular detrital grains with a 179 grain size distribution, the clay as a stringer, the near-irreducible water as covering the quartz grain surfaces and the oil as 180 occupying the centre parts of the pores as these geometries can be found in typical water-wet shaly sandstone reservoirs. It 181 should be noted, however, that the equations make no such distinction and what follows is true for any geometrical set of 4-182 phases composing the 3D medium completely. Reading downwards, Figure (c), (e), (g) and (i) show each of the quartz, clay, 183 water and oil phases alone and respectively. One can imagine that each phase has a certain phase fraction and a certain 184 connectedness. Some of the phases look disconnected in the figure, but it should be remembered that there will be a greater connectedness in reality because there will be connection in the third dimension that are not shown in the figure. If we imagine 185 186 hydraulic flow or electrical flow from the bottom to the top of the medium, the quartz seems to have a relatively high phase 187 fraction and a moderate connectedness, the clay to have a moderate phase fraction and a high connectedness, the water to have 188 a low phase fraction but a relatively high connectedness due to the multiple pathways formed by the thin 'ribbons' of water, 189 and the oil has a moderate phase fraction, but a relatively low connectedness as the patches of oil are relatively isolated. The 190 right hand part of the figure represents the same medium but with the small addition of a quartz grain, labelled 'Q', and its 191 accompanying thin film of surface water. The addition of this makes a miniscule increase in the phase fractions of the detrital 192 quartz and water phase fractions, and, literally, an equally small decrease in the phase fractions of the clay and oil. Reading 193 the distributions for the quartz, clay, water and oil phases alone (Figure (d), (f), (h) and (j)) shows that the addition has made 194 a significant increase in the connectedness of the quartz as well as some increase in that of the water, which was well connected anyway. The low connectedness of the oil will have changed little, but the addition has blocked the main pathway through the clay, leaving only a minor secondary pathway, and consequently resulting in a significant decrease in the clay connectedness.

Consequently Figure 1 shows the principle behind the idea of the conservation of connectedness given in Equation (4), but not a proof, the latter of which is considered in Glover (2010).

In summary, both the sum of the volume fractions and the sum of the connectednesses of the phases composing a 3D medium is equal to unity. The corollary is that connectedness is conserved; if the connectedness of one phase diminishes, there must be an increase in the connectedness of one or more of the other phases to balance it.

It is interesting to consider the role of percolation effects within the generalized model (see Glover (2010) for a full 203 treatment). In percolation theory, the bulk value of a given transport property is only perturbed by the presence of a given 204 phase with a well-defined phase conductivity after a certain phase volume fraction has been attained. This critical volume 205 fraction is called the percolation threshold. This works well for a two-phase system when one phase is nonconductive, with a percolation threshold occurring near the 0.3316 to 0.342 (Montaron, 2009). For such a system, one non-conducting and one 206 207 conducting phase, the effective conductivity of the medium depends only on the conductivity of the conducting phase, its 208 volume fraction and how connected it is. It is intuitive, therefore, that there may exist a phase volume fraction below which 209 the conducting phase is not connected and for which the resulting effective conductivity will be zero. The concept of a 210 percolation factor becomes unclear if the matrix phase has a non-zero conductivity or one or more additional, either solid or 211 fluid conducting phases are added. Under these circumstances a percolation threshold may not exist. Glover (2010) went 212 further than this claiming that Equation (4) in this work (which is Equation (26) in Glover (2010)) contains enough information 213 to make the explicit inclusion of percolation effects unnecessary.

**214 **4 Origin of the saturation exponent**

Within the framework of the classical Archie's laws it is possible to envisage the cementation exponent as controlling how the porosity is connected within the rock sample volume, and to envisage the saturation exponent as controlling how the water is connected within that porosity. The cementation exponent is defined relative to the total volume of the rock, while the saturation exponent is defined relative to the pore space, which is a subset of the whole rock. This is an important concept for what follows.

The water is one of two phases within the porosity, while that porosity is one of two phases within the rock. Hence, there exists a three phase system to which the generalised Archie's law can be applied. In fact, the generalised Archie's law can be used to show that the saturation exponents arise naturally and have a physical meaning: they are defined in the same way as the phase exponents but are expressed relative to the pore space instead of the whole rock.

By writing the generalized law (Equation 4) for three defined phases; let's say matrix, water and hydrocarbon gas, and assuming that neither the matrix nor the gas is conductive, *i.e.*,  $\sigma_m = 0$  and  $\sigma_h = 0$ , but allowing the pore space to be partially saturated with water such that  $\phi_h \neq 0$ , it is possible to obtain  $\sigma_{eff} = \sigma_f \phi_f^{m_f}$ . This is a re-expression of Equation (4), which is the sum of three terms, one for each phase, two of which are zero because the conductivity of the material which makes up each of those is zero (*i.e.*, the matrix and hydrocarbon). The exponent  $m_f$  is the phase exponent of the fluid phase, which is the only phase contributing to the effective conductivity of the three phase medium. Since  $\phi_h \neq 0$ , the pore space is partially saturated with hydrocarbon and partially saturated with water. It is also possible to write  $\phi_f = \phi S_w$ , and hence obtain

$$\sigma_{eff} = \sigma_f \phi^{m_f} S_w^{m_f} .$$
 (5)

Comparison with the classical Archie's laws, which can be written as  $\sigma_{eff} = \sigma_f \phi^m S_w^n$  (Tiab and Donaldson, 2004) shows 233 structural similarity. However, the exponent  $m_f$  in Equation 5 is expressed relative to the whole rock because it is the phase

- 234 exponent for the fluid that appears in Equation 4. By contrast, although the cementation exponent *m* in the classical first
- Archie's law is expressed relative to the whole rock, the saturation exponent n is related to the pore space which is a subset of the whole rock. The distinction between whether the exponent is expressed relative to the whole rock or relative to a subset of the rock, such as the pore space, can be made easily by imagining whether the saturation exponent is independent of any changes one might make to the rock matrix. In this case it is possible to see that the saturation exponent is independent of the rock matrix, and is only sensitive to changes occurring within the pore space. Consequently, it is expressed relative to the pore space rather than the whole rock.
- Accordingly, both equations provide a valid measure of the effective rock conductivity, so they may be equated as  $\sigma_f \phi^{m_f} S_w^{m_f} = \sigma_{eff} = \sigma_f \phi^m S_w^n$ , hence obtaining  $\phi^m = \phi^{m_f} S_w^{(m_f - n)}$ . It can be recognized that the classical Archie saturation exponent refers to saturation with water and is hence renamed as  $n_f$ , giving
- 244

$$\phi^m = \phi^{m_f} S_w^{(m_f - n_f)} \,. \tag{6}$$

It is important to realize that the exponent  $n_f$  is a 'saturation' exponent that refers to the arrangement of the water phase within the pore space. In other words it is expressed with respect to the pore space, not the whole rock, and is found experimentally by varying the saturation of the water in the pore space, the latter of which is assumed to always remain unchanged.

Now it is possible to write Equation (6) in terms of connectednesses. The left hand side of Equation (6) is simply the connectedness of the pore space, as defined by Equation (1). It is the phase volume fraction of the pore space, i.e., the classical porosity, raised to the power of the phase exponent that contains the information about how that pore space is distributed, which is the classical cementation exponent *m*. Consequently we can write  $G_{pore} = \phi^m$ , and Equation (6) becomes

$$G_{pore} = \phi^{m_f} S_w^{(m_f - n_f)}.$$
(7)

The right hand side of the equation may be rewritten as  $(\phi S_w)^{m_f} / S_w^{n_f}$  which allows Equation (7) to be written as

$$G_{pore}S_w^{n_f} = (\phi S_w)^{m_f}.$$
 (8)

The term in brackets is simply the phase fraction of the water with respect to the whole rock, i.e.,  $\phi_f = \phi S_w$  and the exponent mf is simply the phase exponent of the fluid phase with respect to the whole rock. Consequently, Equation (1) can be applied for the fluid phase leading to

$$G_{f} = \phi_{f}^{m_{f}} = (\phi S_{w})^{m_{f}}, \qquad (9)$$

which, when substituted into Equation (8) and rearranged gives

 $G_f = G_{pore} S_w^{n_f} \,. \tag{10}$

This equation is for one fluid phase, i.e., water, occupying the pore space. Since the system is symmetric Equation (10) can be generalized for any of the fluid phases occupying the pore space

$$G_j = G_{pore} S_j^{n_j}, \tag{11}$$

where  $G_j$  is the connectedness of fluid *j*,  $S_j$  is its saturation, and the exponent  $n_j$  is a 'saturation' exponent that refers to the arrangement of the water phase within the pore space. In other words  $n_j$  is expressed with respect to the pore space, not the whole rock.

However, there is nothing geometrically special about the entity we call the pore space or any distinction between solid and fluid phases that compose the whole rock. Consequently, Equation (11) is only a partial generalization, and it is possible

- 270 to extend the result in Equation (10) to any phase of i phases composing a three-dimensional medium each of which partially
- 271 or fully occupies a saturation  $S_i$  of a subset of the medium whose connectedness is given as  $G_{ref}$ , according to

$$G_i = G_{ref} S_i^{n_i} \,.$$

The pore connectedness is relabelled as the reference connectedness because the equation is valid not only for multiple phases 273 that fill the porosity, but multiple phases composing any other phase. 274

Equation 12 gives the connectedness of the  $i^{th}$  phase in an *n*-phase 3D medium as depending on both its fractional saturation 275  $S_i$  within a larger volume which has a connectedness  $G_{ref}$  and that reference connectedness. The distribution of that saturation 276

is taken into account by the exponent  $n_i$ , which will have a general functional form. 277

If one considers the whole 3D *n*-phase medium (i.e., one where  $\sum_{i} \phi_i = 1$ ), Equation 1 states that the connectedness of 278

each phase is the volume fraction of that phase raised to the value of its phase exponent, and Equation 4 states that the sum of 280 those connectednesses is unity.

If a subset of a whole *n*-phase medium (i.e., one where  $\sum_{i} \phi_i < 1$ ) is considered, and labelled the reference subset, the 281

reference subset will have a connectedness  $G_{ref} = \phi_{ref}^{m_{ref}}$  relative to the whole rock, and the connectedness of any phase which 282 partially occupies the reference subset (e.g., water within the pore space, clay within the rock matrix etc.) is equal to the 283 284 connectedness of the reference phase multiplied by the volume fraction of the phase within the reference subset (i.e., the 285 saturation relative to the reference subset) raised to the value of its saturation exponent.

The definition above is somewhat complex due to the requirement to be both completely general and precise, and that there 286 are two reference frames here. The first is the whole 3D n-phase medium. The second is the 3D reference subset which may 287 contain between 2 and n-1 phases. Conversion between the two reference frames can be carried out using the relationship 288

> $\phi_i^{m_i-n_i} = \phi_{ref}^{m_{ref}-n_i},$ (13)

It can also be shown that (Glover, 2010)

$$\sum_{i} S_i^{n_i} = 1, \tag{14}$$

(12)

(16)

where the sum is carried out over all the phases within the reference subset.

It should be noted that Equation 14 is formally the same as Equation 4 except that Equation 14 is valid for the reference 294 subset of phases, while Equation 4 is valid for the whole *n*-phase medium. Hence it is possible to use  $S_i = \phi_i / \phi_{ref}$  to write 295 both Equation 4 and Equation 14 as

> $\sum_{i=1}^{m_i} \left(\frac{\phi_i}{\phi_{rot}}\right)^{m_i} = 1.$ (15)

For a whole *n*-phase medium  $\phi_{ref} = 1$  and Equation 15 becomes equal to Equation 4.

For a subset of the *n*-phase medium  $\phi_{ref} < 1$  and Equation 15 becomes equal to Equation 14.

The distinction between the phase exponent and saturation exponent becomes trivial; they each control how connected the phase is relative to the reference volume fraction. In other words, the transformation 300

 $1 \leftrightarrow \phi_{ref}$  leading to  $\phi_i \leftrightarrow S_i$  and  $m_i \leftrightarrow n_i$ . 301

Figure 2 illustrates the concept of a subset of an *n*-phase medium using a 2D slice from a 3D medium. Figure 2(a) shows a 303 simple 2 phase situation where Phase 1 is brown and Phase 2 is yellow. Both phase are connected across the medium from top 304 to bottom, and were they not in the 2D slice, they would likely be connected through the third dimension. Phase 1 (brown) can 305 be considered as the solid matrix of a rock, and Phase 2 (yellow) is considered to be the pore spaces in the rock for the purposes of this illustration, but the distinction is arbitrary. The rock matrix has a phase fraction  $\phi_1$  and a connectedness  $G_1 = \phi_1^{m_1}$  and the pore space has a phase fraction  $\phi_2$  and a connectedness  $G_2 = \phi_2^{m_2}$  (Eq. (1)). Both of these are expressed with respect to the whole medium that is bounded in the figure by the dashed box. Consequently,  $\phi_1 + \phi_2 = 1$  and  $G_1 + G_2 = 1$  (Eqs. (3) and (4)). The pore space may be occupied by any number of miscible or immiscible fluids. Let us assume there are 2 immiscible fluids completely occupying the pores, which are water and oil, and which we will assign the names Phase 3 and Phase 4.

Figure 2(b) shows this situation. Once again the phase fraction and connectedness of each of the three phases that compose the medium can be defined as phase fractions  $\phi_1$ ,  $\phi_3$  and  $\phi_4$  and  $G_1 = \phi_1^{m_1}$ ,  $G_3 = \phi_3^{m_3}$  and  $G_4 = \phi_4^{m_4}$  for the solid matrix, water and oil, respectively. Since these parameters are being considered with respect to the whole medium it is possible to write

$$\sum_{i=1,3,4} \phi_i = 1$$
 and  $\sum_{i=1,3,4} G_i = 1$

However, it is possible to use a different reference medium for calculations. For example the classical Archie's second law 316 is expressed in terms of saturations, which uses the pore space as a reference space in order to express the amount of water 317 and hydrocarbons, not with respect to the total volume of the rock, but as a fraction of the pore space. Let us, therefore also 318 take the pore space as a convenient reference sub-space of the whole medium. This situation is shown in Figure 2(c) where the 319 dotted line delineated the extent of the reference space. In this space (i) what was the whole medium, represented by unity in the transform given in Eq. (16) becomes the volume fraction of the reference space  $1 \leftrightarrow \phi_{ref}$  (i.e., the pore space in this 320 321 example, (ii) the volumes of the different phases are more efficiently described using saturations Si with respect to the reference 322 space (i.e., the pore space) than using phase volume fractions which are defined relative to the whole medium  $\phi_i \leftrightarrow S_i$ , and (iii) the whole medium connectednesses  $G_i = \phi_i^{m_i}$  are replaced by the entity  $S_i^{n_i}$ , which uses the saturation exponent in place 323 of the phase exponent  $m_i \leftrightarrow n_i$ . It will be seen that the entity  $S_i^{n_i}$  has its own properties in the next section and will be labelled 324 the fractional connectedness. Topologically, the occupation of the fluids within the pore space (Figure 2(c)) is identical to the 325 326 occupation of the whole medium by the matrix and pore space (Figure 2(a)), which leads to the symmetry in the mathematical 327 equations.

The transformation given in Eq. (16) is perhaps not immediately clear when expressed in these most general terms. Let us 329 take an illustrative example. Imagine a three-dimensional 5-phase medium where the phases are (i) detrital quartz (dq), (ii) 330 calcite cement (cc), (iii) distributed clay (dc), (iv) saline water (sw), and (v) hydrocarbon gas (hg), where the subscripts that will be used for each phase are given in parentheses. First let us consider the whole medium (i.e.,  $\phi_{ref} = 1$ ). Each of the phase 331 332 volume fractions are given by  $\phi_{dq}$ ,  $\phi_{cc}$ ,  $\phi_{dc}$ ,  $\phi_{sw}$ , and  $\phi_{hg}$ , respectively. Each of their connectednesses is equal to their phase 333 volume fraction raised to the power of their phase exponents (according to Equation (1)), where the phase exponents contain 334 the information about how each of the five phases is distributed in the medium. The connectednesses are  $G_{da} = \phi_{da}^{m_{dq}}, G_{cc} = \phi_{cc}^{m_{cc}}, G_{dc} = \phi_{dc}^{m_{dc}}, G_{sw} = \phi_{sw}^{m_{sw}}, G_{hg} = \phi_{hg}^{m_{hg}}$ , respectively. Equation (15) can be used, setting  $\phi_{ref} = 1$ , to give 335

$$\phi_{dq}^{m_{dq}} + \phi_{cc}^{m_{cc}} + \phi_{dc}^{m_{dc}} + \phi_{sw}^{m_{sw}} + \phi_{hg}^{m_{hg}} = 1.$$
(17)

This is the same result as applying Equation (4) directly. It is expressed in terms of the parameters (i)  $\phi_{ref} = 1$  (i.e., the whole medium), (ii) individual phase fractions ( $\phi_i$ ), and (iii) individual phase exponents ( $m_i$ ); the latter two of which are expressed relative to the whole medium. These are the conditions and parameters expressed by the left hand components of the transformation given by Equation (16).

Now consider the subset of the whole medium which comprises just its solid parts. The reference fraction  $\phi_{ref}$  is the sum of the solid phase fractions (i.e.,  $\phi_{dq}^{m_{dq}} + \phi_{cc}^{m_{cc}} + \phi_{dc}^{m_{dc}}$ ), which is less than unity. Rewriting Equation (15) for the reference subset gives

$$\qquad \left(\frac{\phi_{dq}^{m_{dq}}}{\phi_{dq}^{m_{dq}} + \phi_{cc}^{m_{cc}} + \phi_{dc}^{m_{dc}}}\right)^{n_{dq}} + \left(\frac{\phi_{cc}^{m_{cc}}}{\phi_{dq}^{m_{dq}} + \phi_{cc}^{m_{cc}} + \phi_{dc}^{m_{dc}}}\right)^{n_{cc}} + \left(\frac{\phi_{dc}^{m_{dc}}}{\phi_{dq}^{m_{dq}} + \phi_{cc}^{m_{cc}} + \phi_{dc}^{m_{dc}}}\right)^{n_{dc}} = 1,$$
(18)

which can be written in terms of 'saturations' (i.e., fractional volumes of the reference subset) as

$$S_{dq}^{n_{dq}} + S_{cc}^{n_{cc}} + S_{dc}^{n_{dc}} = 1,$$
(19)

because
$$S_{dq} = \phi_{dq}^{m_{dq}} / (\phi_{dq}^{m_{dq}} + \phi_{cc}^{m_{cc}} + \phi_{dc}^{m_{dc}})$$
 etc.

There are two important aspects about Equation (19) to note. First, there are no terms for the saline water and hydrocarbon gas in the equation because these phases are not present in the reference subset. Second, that the phase exponents that were used when considering the whole medium have been replaced by saturation exponents because we are now considering the distribution of each of the phases within the reference subset rather than within the whole medium. Third, both Equation (17) and Equation (19) are simultaneously true and may be equated.

Equation (19) is clearly the same as Equation (14). Under the transformation that considers a subset of the whole medium (in this case the solid fractions only) where  $1 \leftrightarrow \phi_{ref}$  the individual phase fractions rlating to the whole medium are replaced by saturations relative to the subset (i.e.,  $\phi_i \leftrightarrow S_i$ ) and the original phase exponents, which were related to the whole medium are now 'saturation' exponents that are related only to the reference subset (i.e.,  $m_i \leftrightarrow n_i$ ).

Both the phase (cementation) exponent and the saturation exponent control how the phase is connected. The phase exponent does this with reference to the whole rock, while the saturation exponent does it with reference to a subset of the whole rock. The underlying physical meaning of the saturation exponent is the same as that of the phase (cementation) exponent, it is only the reference frame that changes. The implication is that the general Archie's law replaces both of the classical Archie's laws. For an application to a sandstone gas reservoir, one would use a 3 phase generalized Archie law.

Equation 12 is easily transformed to provide a calculable value for the saturation exponent by taking the logarithm of both sides of Equation (12) and rearranging the result before substituting Equation (1) for the relevant connectednesses and using the relationship  $S_i = \phi_i / \phi_{ref}$  to obtain

$$n_{i} = \frac{\log(G_{i}) - \log(G_{ref})}{\log(S_{i})} = \frac{m_{i}\log(\phi_{i}) - m_{ref}\log(\phi_{ref})}{\log(\phi_{i}) - \log(\phi_{ref})}.$$
 (20)

This equation may be illustrated using a three phase medium. Imagine a reservoir rock with a 20% porosity. The pore space contains only oil and water with a water saturation of 0.25. We want to calculate the saturation exponent of the water if the 367 phase exponents of the matrix and the oil are 0.2 and 1.68, respectively. It is simple to calculate the volume fractions of matrix 368 369 oil and water to be 0.8, 0.15 and 0.05, respectively. The connectednesses of matrix and oil can be calculated using Equation 370 (1) to be 0.956 and 0.0413, respectively. Using Equation (4) we obtain the connectednesses of the pores and water to be 0.0436 and 0.00236, respectively. In this case the reference subset is the pore space so  $G_{ref} = G_{pore} = 0.0436$ . Equation (20) can now 371 372 be used with  $G_{water} = 0.00236$ ,  $G_{ref} = 0.0436$  and  $S_w = 0.25$  to give  $n_w = 2.105$ . The saturation exponent of the oil can also be 373 calculated as  $n_0 = 0.1931$ . There is no value for the matrix as the matrix is not included in the pore space reference subset.

There is a reiterative symmetry in this transformation where both the whole medium phase fractions and the reference subset saturations are both volume fractions with respect to the whole medium and the reference subset, respectively. Similarly, the phase exponents and the saturation exponents are also defined with respect to the whole medium and the reference subset, respectively. This would, therefore, allow the calculation of a reference subset of a subset of a whole medium if required, and so on. There is of course the possibility that the whole *n*-phase medium is itself a subset of a larger medium with more phases. In this case Equation 15 still holds, but with  $\phi_{ref} > 1$ . The implication is that the definition of the original whole medium is arbitrary and can be defined to make the solution of the problem more tractable.

**381 **5** Physical interpretation of the saturation exponent**

This section provides a physical interpretation for the saturation exponent in a perfect analogy to that derived for the cementation exponent by Glover (2009).

The connectedness *G* is the inverse of the Archie's formation factor and is central to the generalized Archie's law. The inverse of the Archie's resistivity (saturation) index  $1/I = S_w^n$  is also rather important. It relates the connectednesses of each phase with respect to the whole rock to the connectedness of the reference subset in Equation 12, and when summed over all the phases that occupy the reference subset it produces unity as in Equation 14. In this paper the inverse of the Archie's resistivity (saturation) index has been given the symbol  $H_i$  and defined as

$$H_i \equiv S_i^{n_i} \,. \tag{21}$$

Just as the saturation of any given phase  $S_i$  is the ratio of the volume fraction of the phase to that of the all the phases making up any reference set of phases,  $H_i$  is the ratio of the connectedness of the phase to that of the all the phases making up any reference set of phases. The parameter  $H_i$  is in fact a fractional connectedness.

We follow the approach of Glover (2009) in the analysis of the physical interpretation of the cementation exponent. In this work Glover (2009) showed that the cementation exponent was the differential of the connectedness with respect to both porosity and pore connectivity. Following the same methodology, differentiating the fractional connectedness with respect to the phase saturation  $S_i$  gives

$$\frac{\partial H_i}{\partial S} = n_i S_i^{n_i - 1} \,. \tag{22}$$

By analogy we recognize that  $S_i^{n,-1}$  represents the connectivity of Phase *i* with respect to the reference subset and define this connectivity as

$$\psi_i = S_i^{n_i - 1},\tag{23}$$

to give

$$\frac{\partial H_i}{\partial S} = n_i \psi_i \,. \tag{24}$$

A further differentiation, this time with respect to the connectivity  $\psi_i$  allows us to obtain

$$n_i = \frac{\partial}{\partial \psi} \left( \frac{\partial H_i}{\partial S} \right).$$
 (25)

Consequently, the saturation exponent is the rate of change of fractional connectedness with respect to both phase saturation and phase connectivity in a similar way that Glover (2009) found that the physical interpretation of the cementation exponent was the rate of change of connectedness with respect to phase fraction (porosity) and its connectivity. This shows once again the symmetry between phase fractions and saturations and between phase exponents and saturation exponents.

The fractional connectedness is also the product of the saturation and the connectivity with respect to the reference subset 410  $H_i = S_i \psi_i$ . (26)

Hence, the saturation exponents obey the same laws as the phase (cementation) exponents, but whereas the phase exponents are defined relative to the whole rock, the saturation exponents are defined relative to some subset of the rock. Table 1 shows the relationships of the generalised Archie's law expressed relative to the whole rock and with respect to a reference subset of the whole rock.

For petrophysicists the reference subset has been the porosity and there has only been one conducting phase that partially saturates that porosity – the pore water. Now we are not restricted to that model. The reference subset could be, for example, the solid matrix in which a number of separate mineral phases can be defined, one of which might be, say, a target ore or a clay phase. Let us take a 4-phase medium as an example. Imagine a 4 phase medium composed of 65% quartz matrix with a 419 phase volume exponent of 0.3, 15% clay. Consequently, the medium's porosity is  $\phi=0.2$ . The porosity is occupied with gas and saline water with saturations  $S_g=0.625$  and  $S_w=0.375$ , respectively and the classical commutation exponent m=1.8 and 420 the classical saturation exponent is n = 2.05. Imagine needing to calculate the resistivity of the rock if the resistivity of the clay 421 and the water are known,  $\rho_{clay} = 50 \ \Omega$ .m and  $\rho_{water} = 5 \ \Omega$ .m, say. Equation (1) can be used to calculate  $G_{quartz} = 0.8788$  and 422  $G_{pore} = 0.0552$ . Using Equation (4) provides  $G_{clay} = 0.0660$  with no need to consider the various saturations of the fluids 423 424 occupying the pores. The phase exponent of the clay can be found to be  $m_{clay}=1.43$ . The contribution of the clay to resistivity can be calculated using Equation (3), rewritten as  $\rho_{contclay} = \rho_{clay} \phi_{clay}^{-m_{clay}} = \rho_{clay} / G_{clay}$ , as  $\rho_{clay} = 757 \Omega$ .m, noting that this value 425 426 takes full account of its volume fraction and its geometrical distribution. Now we must consider the relative distributions of 427 water and gas in the medium. Calculations can be carried out in terms of connectednesses G or fractional connectednesses H. In this case we use the connectednesses G. Equation (11) or (12) can be used to calculate  $G_{water}$ =0.00739, and Equation (4) 428 applied to give  $G_{gas} = 0.0478$ . Once again Equation (1) may be applied, but this time in the rearranged form  $m_i = \log G_i / \log \phi_i$ 429 in order to calculate the respective phase exponents  $m_{water} = 1.895$  and  $m_{gas} = 1.462$ . Now, the contribution of the saline water 430 to the overall resistivity can be calculated using Equation (3), rewritten as  $\rho_{contcwater} = \rho_{water} \phi_{water}^{-m_{water}} = \rho_{water} / G_{water}$ , as  $\rho_{water} = \rho_{water} / G_{water}$ 431 432 677 Ω.m, noting that this value takes full account of its volume fraction and its geometrical distribution. The resistivity of the 433 rock can now be calculated by simply summing the contributions to conductivity as implied by Equation (3) to give  $\rho_{eff} = 357$  $\Omega$ .m. In this particular example, the conductivity of the medium is controlled by the clay and water fractions in approximately 434 equal measure. It should also be noted that there are a number of different pathways for obtaining the same result using the 435 436 equations contained in this paper.

**437 6 Conclusions**

The main conceptual steps in this paper are summarised as:

- The classical Archie saturation exponent arises naturally from the generalised Archie's law.
- The saturation exponent of any given phase can be thought of as formally the same as the phase (i.e., cementation) 441 exponent, but with respect to a reference subset of phases in a larger *n*-phase medium.
- The connectedness of each of the phases occupying a reference subset of an *n*-phase medium can be related to the 443 connectedness of the subset itself by  $G_i = G_{ref} S_i^{n_i}$ .
- The sum of the connectednesses of a 3D *n*-phase medium is given by  $\sum_{i} \phi_{i}^{m_{i}} = 1$ , mirroring the relationship for phase volumes
$$\sum_{i} \phi_i = 1$$
.

Connectedness is conserved in a 3D *n*-phase medium. If one phase increases in connectedness the connectedness of one
 or more of the other phases must decrease to compensate for it, just as phase volumes are conserved with the decrease in
 one leading to the increase of another phase.

- The sum of the fractional connectednesses (saturations) of an *n*-phase medium is given by  $\sum_{i} S_{i}^{n_{i}} = 1$ .
- Fractional connectedness is conserved in a 3D *n*-phase medium.
- 451 The saturation exponent may be calculated using the relationship  $n_i = \frac{m_i \log(\phi_i) m_{ref} \log(\phi_{ref})}{\log(\phi_i) \log(\phi_{ref})}$ .
- The connectivity of any phase with respect to the reference subset is given by  $\psi_i = S_i^{n_i-1}$ .

- The connectedness of a phase with respect to a reference subset (also called the fractional connectedness) is given by 454  $H_i = S_i \psi_i$  and depends upon the fractional volume of the phase divided by that of the reference subset (i.e., its saturation)
- and the arrangement of the phase within the reference subset (i.e., its connectivity with respect to the reference subset).

• The rate of change of fractional connectedness with saturation  $\frac{dH_i}{dS_i} = n_i \psi_i$  depends upon the connectivity with respect

- 457 to the reference subset  $\psi_i$  and the saturation exponent  $n_i$ .
- 458 Hence, the saturation exponent is interpreted as being the rate of change of the fractional connectedness with saturation and connectivity within the reference subset,  $n_i = \frac{d^2 H_i}{d\psi_i dS_i}$ .

While this paper represents a theoretical treatment of the saturation exponent and attempts to develop a theoretical interpretation that should offer insight into the physical meaning of the saturation exponent, it does not contain a physical
proof of these equations. That can only come from targeted experimental work on multi-phase media which are difficult to
carry out and represent one of our research goals.

| 465 | Table 1. | Comparison | of all the p | parameters in | n the classical | and general | ised Archie's laws. |
|-----|----------|------------|--------------|---------------|-----------------|-------------|---------------------|
|-----|----------|------------|--------------|---------------|-----------------|-------------|---------------------|

| Parameter                                  | Generalis                                                                                                                                                                                                                                                                                                                                                                                                                                                                                                                                                                                                                                                                                                                                                                                                                                                                                                                                                                                                                                                                                                                                                                                                                                                                                                                                                                                                                                                                                                                                                                                                                                                                                                                                                                                                                                                                                                                                                                                                                                                                                                                                                                                                                                                                                                                                                                                                                                                                                                                                                                                                                                                                                                                                                                                                                                                                                                                                                                                                                                                                                                                                                                         | sed Archie's law                                                                                                                                                                                       | Classical Archie's law                                         |                                                                 |  |
|--------------------------------------------|-----------------------------------------------------------------------------------------------------------------------------------------------------------------------------------------------------------------------------------------------------------------------------------------------------------------------------------------------------------------------------------------------------------------------------------------------------------------------------------------------------------------------------------------------------------------------------------------------------------------------------------------------------------------------------------------------------------------------------------------------------------------------------------------------------------------------------------------------------------------------------------------------------------------------------------------------------------------------------------------------------------------------------------------------------------------------------------------------------------------------------------------------------------------------------------------------------------------------------------------------------------------------------------------------------------------------------------------------------------------------------------------------------------------------------------------------------------------------------------------------------------------------------------------------------------------------------------------------------------------------------------------------------------------------------------------------------------------------------------------------------------------------------------------------------------------------------------------------------------------------------------------------------------------------------------------------------------------------------------------------------------------------------------------------------------------------------------------------------------------------------------------------------------------------------------------------------------------------------------------------------------------------------------------------------------------------------------------------------------------------------------------------------------------------------------------------------------------------------------------------------------------------------------------------------------------------------------------------------------------------------------------------------------------------------------------------------------------------------------------------------------------------------------------------------------------------------------------------------------------------------------------------------------------------------------------------------------------------------------------------------------------------------------------------------------------------------------------------------------------------------------------------------------------------------------|--------------------------------------------------------------------------------------------------------------------------------------------------------------------------------------------------------|----------------------------------------------------------------|-----------------------------------------------------------------|--|
|                                            | With respect to the whole medium                                                                                                                                                                                                                                                                                                                                                                                                                                                                                                                                                                                                                                                                                                                                                                                                                                                                                                                                                                                                                                                                                                                                                                                                                                                                                                                                                                                                                                                                                                                                                                                                                                                                                                                                                                                                                                                                                                                                                                                                                                                                                                                                                                                                                                                                                                                                                                                                                                                                                                                                                                                                                                                                                                                                                                                                                                                                                                                                                                                                                                                                                                                                                  | With respect to a reference subset of the whole medium                                                                                                                                                 | First law                                                      | Second Law                                                      |  |
| Phase
volume
fraction                | $egin{aligned} eta_i \ \phi_i &= \phi_{ref} \ S_i \end{aligned}$                                                                                                                                                                                                                                                                                                                                                                                                                                                                                                                                                                                                                                                                                                                                                                                                                                                                                                                                                                                                                                                                                                                                                                                                                                                                                                                                                                                                                                                                                                                                                                                                                                                                                                                                                                                                                                                                                                                                                                                                                                                                                                                                                                                                                                                                                                                                                                                                                                                                                                                                                                                                                                                                                                                                                                                                                                                                                                                                                                                                                                                                                                                  | $S_i \ S_i = \phi_i / \phi_{ref}$                                                                                                                                                                      | $\phi \\ V_f = V_{pore}  S_w$                                  | $S_{w} = V_{f} / V_{pore}$                                      |  |
| Exponent                                   | $m_{i} = \frac{d}{d\chi} \left( \frac{dG_{i}}{d\phi} \right)$ $m_{i} = \frac{\log(\sigma_{i}) - \log(\sigma_{f})}{\log(\phi_{i})}$                                                                                                                                                                                                                                                                                                                                                                                                                                                                                                                                                                                                                                                                                                                                                                                                                                                                                                                                                                                                                                                                                                                                                                                                                                                                                                                                                                                                                                                                                                                                                                                                                                                                                                                                                                                                                                                                                                                                                                                                                                                                                                                                                                                                                                                                                                                                                                                                                                                                                                                                                                                                                                                                                                                                                                                                                                                                                                                                                                                                                                                | $\begin{split} n_{i} = & \frac{d}{d\psi} \left( \frac{dH_{i}}{dS} \right) \\ n_{i} = & \frac{m_{i}\log(\phi_{i}) - m_{ref}\log(\phi_{ref})}{\log(\phi_{i}) - \log(\phi_{ref})} \end{split}$            | $m = \frac{\log(\sigma_{eff}) - \log(\sigma_{f})}{\log(\phi)}$ | $n = \frac{\log(\sigma_{eff}) - \log(\sigma_{100})}{\log(S_w)}$ |  |
| Connected-
ness                         | $ \begin{array}{ccc} G_i \equiv \phi_i^{m_i} & H_i \equiv S_i^{n_i} \\ G_i = \phi_i \chi_i & H_i = S_i \psi_i \\ G_i = 1/F_i & H_i = 1/I_i \\ G_i = G_{ref} H_i & H_i = G_i/G_{ref} \end{array} $                                                                                                                                                                                                                                                                                                                                                                                                                                                                                                                                                                                                                                                                                                                                                                                                                                                                                                                                                                                                                                                                                                                                                                                                                                                                                                                                                                                                                                                                                                                                                                                                                                                                                                                                                                                                                                                                                                                                                                                                                                                                                                                                                                                                                                                                                                                                                                                                                                                                                                                                                                                                                                                                                                                                                                                                                                                                                                                                                                                 |                                                                                                                                                                                                        | undefined                                                      | undefined                                                       |  |
| Connect-
ivity                          | $\chi=\phi_i^{m_i-1}$                                                                                                                                                                                                                                                                                                                                                                                                                                                                                                                                                                                                                                                                                                                                                                                                                                                                                                                                                                                                                                                                                                                                                                                                                                                                                                                                                                                                                                                                                                                                                                                                                                                                                                                                                                                                                                                                                                                                                                                                                                                                                                                                                                                                                                                                                                                                                                                                                                                                                                                                                                                                                                                                                                                                                                                                                                                                                                                                                                                                                                                                                                                                                             | $\psi = S_i^{n_i - 1}$                                                                                                                                                                                 | $\chi = \phi^{m - 1}$                                          | undefined                                                       |  |
| Rate of
change of
connected-
ness | $\frac{dG_i}{d\phi_i} = m_i \chi_i$                                                                                                                                                                                                                                                                                                                                                                                                                                                                                                                                                                                                                                                                                                                                                                                                                                                                                                                                                                                                                                                                                                                                                                                                                                                                                                                                                                                                                                                                                                                                                                                                                                                                                                                                                                                                                                                                                                                                                                                                                                                                                                                                                                                                                                                                                                                                                                                                                                                                                                                                                                                                                                                                                                                                                                                                                                                                                                                                                                                                                                                                                                                                               | $\frac{dH_i}{dS_i} = n_i \psi_i$                                                                                                                                                                       | undefined                                                      | undefined                                                       |  |
| Sum of
phases                           | $\sum_{i=1}^{i} \phi_i = 1$ $\sum_i S_i > 1$                                                                                                                                                                                                                                                                                                                                                                                                                                                                                                                                                                                                                                                                                                                                                                                                                                                                                                                                                                                                                                                                                                                                                                                                                                                                                                                                                                                                                                                                                                                                                                                                                                                                                                                                                                                                                                                                                                                                                                                                                                                                                                                                                                                                                                                                                                                                                                                                                                                                                                                                                                                                                                                                                                                                                                                                                                                                                                                                                                                                                                                                                                                                      | $\sum_{i=1}^{i} \phi_i < 1$ $\sum_{i} S_i = 1$                                                                                                                                                         | $\phi_{pore} + \phi_{matrix} = 1$                              | $S_w + S_o + S_g = 1$                                           |  |
| Sum of
connected-
nesses             | $\sum_{i} \phi_{i}^{m_{i}} = \sum_{i} G_{i} = 1$ $\sum_{i} \int_{i} $ | $\sum_{i} S_{i}^{n_{i}} = \sum_{i} H_{i} = 1$ $\left(\frac{\phi_{i}}{\phi_{ref}}\right)^{m_{i}} = 1$ $e_{f} \text{ leads to } \phi_{i} \leftrightarrow S_{i} \text{ and } m_{i} \leftrightarrow n_{i}$ | undefined                                                      | undefined                                                       |  |
| Effective
conduct-
ivity             | $\sigma_{eff} = \sum_{i} \sigma_{i} \phi_{i}^{m_{i}}$                                                                                                                                                                                                                                                                                                                                                                                                                                                                                                                                                                                                                                                                                                                                                                                                                                                                                                                                                                                                                                                                                                                                                                                                                                                                                                                                                                                                                                                                                                                                                                                                                                                                                                                                                                                                                                                                                                                                                                                                                                                                                                                                                                                                                                                                                                                                                                                                                                                                                                                                                                                                                                                                                                                                                                                                                                                                                                                                                                                                                                                                                                                             | $\sigma_{e\!f\!f} = \sum_i \sigma_i \phi^{m_i}_{ref} S^{m_i}_i$                                                                                                                                        | $\sigma_{e\!f\!f}=\sigma_{f}\phi^{m}$                          | $\sigma_{eff} = \sigma_f \phi^m S_w^n$                          |  |

**Data availability**

This work is entirely theoretical and contains no data.

---

## Author Comment (AC2) · 9 May 2017

**Author's Response to Reviewer's Comments for Solid Earth SE-2017-5**

**by Paul Glover**

This document is structured in the following way. There are 3 reviewers who submitted their reviews chronologically. Each one is treated in turn, first quoting the comments of the reviewer, and then responding to them.

**Reviewer 1's comments (Harald Milsch)**

**Review on paper manuscript se-2017-5**

**Summary**
In this paper and based on earlier findings (Glover, 2009; 2010), the author derives a new theoretical interpretation of the saturation index contained in Archie's second law. The essence of this interpretation is the extension of the "generalized Archie's law" outlined by the author in Glover (2010), where the saturation index is viewed as being "formally the same as the phase exponent, but with respect to a reference subset of phases in a larger n-phase medium".

The author carried out an important task with implications for fundamental rock physics and industrial applications alike. The paper is well structured and, in my perception, mathematically sound and may definitely be suitable for publication in Solid Earth (SE).

However, there are a number of substantial issues outlined in the following that I encourage the author to address before the paper can be recommended for publication.

**General comments**
1. It should be noted that (1) what is attempted here is a physical interpretation of an empirical parameter, which I find per se problematic. Also, it should be noted that (2) the outlined interpretation comes as an ad hoc approach and that (3) no proof is presented that this approach and the resulting interpretation is physically correct. Please comment and clarify within the manuscript.

2. The motivation for performing this particular theoretical investigation is well presented in Section 1. However, this discussion also implicitly suggests that reserves calculations can now be performed with unprecedented precision. There is no proof that this is the case. It should also be noted that there will still be experiments and/or analyses to be performed to parameterize the newly introduced equations. How these experiments/analyses should look like and what type of data is required should be included in the text.

3. The theoretical approach is, mathematically, not very demanding but it appears abstract and hard to grasp. I therefore would wish to see (1) some of the equations to be developed in more detail (e.g. in some appendix), (2) one or more graphical representations of the model to better depict the theory, and, not least, (3) a few example calculations where for some type of rock with some kind(s) of fluid(s) some saturation index is derived and then is compared to existing (experimental) data. Please see also comments below.

**Specific comments:**

- Section 3; Lines 148- 152: I wonder if this is correct. What about percolation or a percolation threshold? Please comment. This comment also applies to Line 260.
- Section 3: In this section a first illustrating sketch should be introduced.
- Section 4; Line 171: Please clarify from where this equation arises.
- Section 4; Lines 178-180: Reasoning unclear. Please improve.
- Section 4; Lines 182-185 and 202: The equations contained here should be fully derived, e.g. within the section or some appendix.
- Section 4; Lines 199 and following: Here, a second illustrating sketch should be introduced.
- Section 4; Lines 213-214: Can this transformation be exemplified or illustrated?
- Section 4; Eq. (10): This equation should be fully derived and also (numerically) exemplified for a 3-phase medium like the one mentioned before in Line 221.
- Section 5: The motivation for this section is somewhat unclear and should be outlined.
- Section 5; Line 234: Please briefly recall the approach of Glover (2009).
- Section 5; Eq. (12), (13), and (14): In my opinion the derivation should be improved/expanded and also inverted such that Eq. (12) is the final outcome (as in Section 6).
- Section 5; Eq. (12): This equation is only correct if one can assume that $n_i \neq f(\Psi_i)$. Please show that this is the case.
- Section 5; Eq. (12): Please show that Eq. (12) yields Eq. (10) or vice versa.
- Section 5; Lines 248-250: To illustrate this statement and by applying either equation I would wish to see an example calculation / numerical evaluation for a 4-phase porous medium (e.g. quartz, clay, water, gas).

**Technical corrections**

- The expression "rate of change" suggests some time dependence/derivative and should be replaced throughout the manuscript including in Table 1 by some other, more appropriate, expression.
- Lines 51-52: please check if statement is correct.
- Glover (2016) not in reference list.
- The use of "$\phi$" (phi) for both porosity and phase volume fractions may lead to confusion. Please reconsider.
- Line 125: Equation 4 (?), please check. If correct move Eq. (4) in Line 115 up in text.
- Line 131: please check indices in equation.
- Line 191: Equation 1 (?), please check.
- Line 206: Equation 7 (?), please check.
- Lines 237-238: Index "i" missing in "$\Psi$" (psi).

**References:**

Glover, P. W. J.: What is the cementation exponent? A new interpretation, The Leading Edge, 82–85, doi: 10.1190/1.3064150, 2009.

Glover, P. W. J.: A generalised Archie's law for n phases, Geophysics, 75(6), E247-E265, doi: 10.1190/1.3509781, 2010.

**Author's response Reviewer 1's comments**

**General comments**

| Reviewer's comment | Author's response |
|---|---|
| 1. It should be noted that (1) what is attempted here is a physical interpretation of an empirical parameter, which I find per se problematic … | *Point (1): I disagree with Harald on a philosophical level. I believe that we should ALWAYS seek to find physical interpretations for empirical parameters since (i) it helps understand what they represent, and (ii) it may lead to a deeper understanding of the fundamental theoretical basis underlying the experimental science. It should be noted that most of classical theoretical physics began as an interpretation of empirical observations.* |
| … Also, it should be noted that (2) the outlined interpretation comes as an ad hoc approach and that … | *Point (2): I agree with Harald that the original MS did not contain a sufficiently robust set of derivations and could have been perceived as ad hoc. This was due to me not recognising that my familiarity with manipulations drove me to cut corners in derivations, not realising that most of my colleagues who are not familiar with these equations due to their novelty would need all of the steps to be explicit – they now are!*

*I have heeded the advice of the reviewer and expanded the derivations considerably. There are now 11 extra display equations (an increase of 79%) and numerous extra in-line equations. The associated descriptive text has added about 3664 words, an increase of 73% in the length of the paper, as well as two requested figures with considerable additional explanatory text. The derivations are now extremely robust and clear. Consequently, the paper now represents a theoretical proof of the equations it contains.* |
| … (3) no proof is presented that this approach and the resulting interpretation is physically correct. Please comment and clarify within the manuscript. | *Point (3): Whereas the last point considered the theoretical proof of the equations in the paper, which is now explicit in the revised MS, this point considers the physical proof. That can only come from targeted experimental work, which we are currently asking for funding to carry out, but could also be carried out by colleagues who read this new interpretation. This has now been clarified within the revised MS.* |
| 2. The motivation for performing this particular theoretical investigation is well presented in Section 1. However, this discussion also implicitly suggests that reserves calculations can now be performed with unprecedented precision. There is no proof that this is the case. | *A sentence of 52 words has been added to the relevant section to make this point clear.* |
| It should also be noted that there will still be experiments and/or analyses to be performed | *The need for experimental measurements has now been covered in the conclusions of the MS,* |

| | |
|---|---|
| to parameterize the newly introduced equations. How these experiments/analyses should look like and what type of data is required should be included in the text. | *and has already been commented upon in the reviewer's point above (Point (3)). The detail of the parameters and methodology for such experiments is reserved for the appropriate follow-up paper.* |
| 3. The theoretical approach is, mathematically, not very demanding but it appears abstract and hard to grasp. I therefore would wish to see (1) some of the equations to be developed in more detail (e.g. in some appendix), … | *Harald is perfectly right here. The maths is not complex but the concepts are rather harder to grasp. This point has already been covered in the response to the first general comment above. However, to reiterate: Point (1): This has been done in the text rather than in an appendix resulting in 11 extra display equations (an increase of 79%) and numerous extra in-line equations as well as associated extra descriptive text to clarify some of the difficult conceptual jumps (about 3664 words, an increase of 73% in the length of the paper).* |
| (2) one or more graphical representations of the model to better depict the theory, | *Point (2): Two graphics have been added with over 1000 words of explanatory text.* |
| … and, not least, (3) a few example calculations where for some type of rock with some kind(s) of fluid(s) some saturation index is derived and then is compared to existing (experimental) data. Please see also comments below. | *Point (3): The paper now contains 4 separate example calculations; a 2-phase, a 3-phase, a 4-phase and a 5-phase example, at various points in the paper.* |

**Specific comments:**

| Reviewer's comment | Author's response |
|---|---|
| Section 3; Lines 148- 152: I wonder if this is correct. What about percolation or a percolation threshold? Please comment. | *This comment also applies to Line 260. I see Harald's point here but the problem is already known and considered at length in Glover (2010). In rereading that treatment, I find that I have nothing to add to it and cannot find a better way to say what I said concerning percolation thresholds in that paper. Consequently, I have included a short paragraph (222 words) near the first statement in order to discuss the percolation problem, including an exhortation for the reader to read the relevant parts Glover (2010) if, for them, the issue needs further clarification.* |
| Section 3: In this section a first illustrating sketch should be introduced. | *A new figure has been included, together with 521 words of explanatory text.* |
| Section 4; Line 171: Please clarify from where this equation arises. | *63 words of clarification added.* |
| Section 4; Lines 178-180: Reasoning unclear. Please improve. | *47 words of clarification have been added to or modified the existing text.* |
| Section 4; Lines 182-185 and 202: The equations contained here should be fully derived, e.g. within the section or some appendix. | *A full step-by step derivation has been added in the text, amounting to a significant addition of text and 6 display equations. The mathematics is simple but some of the conceptual steps were not. Hence, I thank the reviewer for flagging up why this derivation would not be understood easily as it was originally stated.* |

| | |
|---|---|
| Section 4; Lines 199 and following: Here, a second illustrating sketch should be introduced. | *A new figure has been included, together with an extra 525 to describe the figure as clearly as possible.* |
| Section 4; Lines 213-214: Can this transformation be exemplified or illustrated? | *A five phase medium has been added as an illustration, taking 497 additional words and 3 display equations.* |
| Section 4; Eq. (10): This equation should be fully derived and also (numerically) exemplified for a 3-phase medium like the one mentioned before in Line 221. | *This is a very simple mathematical manipulation for someone au fait with logarithms. Consequently only 27 words were added in order to explain how the equation is developed step-by-step. No extra equations were necessary. A worked example has been given (184 additional words).* |
| Section 5: The motivation for this section is somewhat unclear and should be outlined. | *30 words have been added including a change of section heading.* |
| Section 5; Line 234: Please briefly recall the approach of Glover (2009). | *This has been done in 48 additional words.* |
| Section 5; Eq. (12), (13), and (14): In my opinion the derivation should be improved/expanded and also inverted such that Eq. (12) is the final outcome (as in Section 6). | *This has been carried out with inversion and the addition of an extra step.* |
| Section 5; Eq. (12): This equation is only correct if one can assume that $n_i \neq f(\psi_i)$. Please show that this is the case. | *This comment is not correct and, consequently, I have made no further changes. The new Eq. (23) shows that $n_i$ is a function of $\psi_i$, but since the differential is with respect to $\psi$ this is a hidden functionality that does not invalidate the use of differentiation nor the resulting differential equation. This mathematical nicety is now clearer thanks to the reordering the equations implemented at the request of the reviewer in the last comment.* |
| Section 5; Eq. (12): Please show that Eq. (12) yields Eq. (10) or vice versa. | *A proof has not been inserted because it is already implicit in the paper. In the new numbering Eqs. (10) and (12) are now Eqs. (20) and (25). Taking each separately, Eq. (20) is derived from Eqs. (6) to (11), which produces Eq. (12) from which Eq. (20) is derived). This process is now much more explicit than the original MS resulting from a response to the reviewer's previous comment, and which has led to a significant improvement in the MS. Eq. (25) is derived in the paper explicitly in Eqs. (21) to (24). Consequently, taking these two explicit derivations together gives the proof for which the reviewer is looking. It is 13 display equations long, and given that it is already in the paper I take the view that to repeat it would be unnecessary.* |
| Section 5; Lines 248-250: To illustrate this statement and by applying either equation I would wish to see an example calculation / | *A 4-phase example has been added (380 words), which in my view shows the power of the equations well. I am grateful to the reviewer for suggesting their inclusion. The paper now* |

| numerical evaluation for a 4-phase porous medium (e.g. quartz, clay, water, gas). | *contains 4 separate example calculations; a 2-phase, a 3-phase, a 4-phase and a 5-phase example, at various points in the paper.* |
| --- | --- |

**Technical corrections**

| Reviewer's comment | Author's response |
| --- | --- |
| The expression "rate of change" suggests some time dependence/derivative and should be replaced throughout the manuscript including in Table 1 by some other, more appropriate, expression. | *"Rate of change" is mentioned in the original manuscript 5 times. I disagree with the reviewer on this point. In the first place I do not consider that a "rate of change" need necessarily imply some time dependence, i.e., a rate of change with respect to time, even if the reference variable is not explicit. However, in this paper the rate of change is given explicitly with respect to either saturation or connectedness in all 5 mentions, and therefore the mathematics is very clearly described. I have not made changes because I believe the reviewer has a view of what "rate of change" means which is more restrictive than the general use.* |
| Lines 51-52: please check if statement is correct. | *Whoops! I gave the derivation for the cementation exponent by mistake. I have changed it so it is an accurate. Many thanks to the reviewer.* |
| Glover (2016) not in reference list . | *Now inserted into the reference list.* |
| the use of "$\phi$" (phi) for both porosity and phase volume fractions may lead to confusion. Please reconsider. | *I had a long think about this. The trouble is that porosity is a phase volume fraction and so using 2 different symbols would make an artificial distinction which is not real. I think that there is not such a difficulty with leading the reader into confusion because the classical porosity is only used up until Equation 2 and then the phase volume fraction terminology takes over, generalising phase volume fractions and incorporating porosity into that structure. In order to avoid confusion I have added 95 words of clarification early in Section 3 explaining the retention of $\phi$ for porosity and the additional use of $\phi_i$ for phase volume fractions. So in the modified form, porosity is considered and used using the symbol $\phi$ up until line 111, and during this time there is no mention of phase volume fractions. Phase volume fractions $\phi_i$ are defined on Line 111 and are used exclusively for the rest of the paper. Consequently I think that the process of generalising porosity into phase volume fractions is carried out in a smooth and clear fashion.* |
| Line 125: Equation 4 (?), please check. If correct move Eq. (4) in Line 115 up in text. | *Corrected. Equation 4, which was actually on line 155 has been moved to line 125 in the* |

| | *original numbering scheme. This is so that references to both equations 1 and 4 that occurred in line 125 do not have to look ahead. This problem was created by adding the 2 phase system example after the rest of the paper have been written, and adding it a little too early.* |
|---|---|
| Line 131: please check indices in equation. | *One subscript corrected.* |
| Line 191: Equation 1 (?), please check. | *This is correct, but I have modified the sentence to make it clearer.* |
| Line 206: Equation 7 (?), please check. | *This should read Equation 8, and has been corrected.* |
| Lines 237-238: Index "i" missing in "Ψ" (psi). | *This has been corrected.* |

**References**
*Both of the suggested references were already listed in the original submission.*
* * *
**Referee 2's comments**

The classical Archie's law is an important expression to describe the relationship between electrical property and the porosity of rocks. In this paper, the author builds a new theory to extend Archie's law and make it more completely, so that it can be applied to the n-phase medium. Although there is a Table listed to make a comparison of these two theories, I do not think it is enough to show the advances and validities of the new theory. It is better to give a simulated analysis in the paper at least. However, the author has presented that there is no data here, so hope to see the related paper soon, which interests me the best.

**Author's response Reviewer 2's comments**

*The reviewer's only substantive comment/wish is to see a simulated analysis. The changes and the three extra examples made in the response to other reviewers, particularly Reviewer 1, should satisfy this.*
* * *
**Referee 3's comments (Graham Heinson)**

An interesting and worthwhile paper on the importance and calculation of Archie's Law saturation exponent. I have little background in this area of petrophysics other than accepting the well-known and simple empirical relationship between resistivity, pore fluid, porosity and saturation. Glover explains both the mathematics in a careful manner and also the context for developing such a theoretical approach. The argument about estimated reserves is both dramatic and perhaps a bit ambit, but it does provide a good reason why a redefinition might matter. Of course, anything connected with such large reserves and value will have a significant effect as a small percentage.

- Line 29 starting "Since..." seems to be missing part of the sentence.

- The sentence from Line 82 - 86 is quite long and could be re-phrased.

- The flow of logic is reasonably well presented. I'm a bit confused on Line 125 that Equation 1 and 4 are mentioned, but Equation 4 does not get defined until line 155. The example for a two-phase system from Line 130 is good in highlighting a simple case.

- The sentence on Line 176 "By contrast..." could be rephrased. My take is that the exponent is related to the fractional volume of pores filled with the fluid rather than being a related to the whole rock. There is a bit of confusing sentence structure.

The conclusions are a nice summary of the paper, but need the paper to make sense of the equations. Thus, they could not really be read stand-alone. Not sure if this is a problem.

**Author's response Reviewer 3's comments**

*The reviewer requires no general or substantial changes, modifications or additions. However, all his specific comments have been acted upon and listed below:*

| Reviewer's comment | Author's response |
|---|---|
| Line 29 starting "Since..." seems to be missing part of the sentence. | *The sentence was originally part of the following sentence and were accidentally separated during editing. The 2 sentences have now been combined again into a correct form.* |
| The sentence from Line 82 - 86 is quite long and could be re-phrased: | *The sentence has been split into at least 4 sentences, and now treats each concept separately.* |
| The flow of logic is reasonably well presented. I'm a bit confused on Line 125 that Equation 1 and 4 are mentioned, but Equation 4 does not get defined until line 155. The example for a two-phase system from Line 130 is good in highlighting a simple case. | *This problem arose because "the 2 phase system example" that the reviewer likes was a late addition that was placed too early. Consequently, Equation 4 has been moved forwards to ensure that logical steps are retained.* |
| The sentence on Line 176 "By contrast..." could be rephrased. My take is that the exponent is related to the fractional volume of pores filled with the fluid rather than being a related to the whole rock. There is a bit of confusing sentence structure: | *The sentence has been rephrased using some of the terminology suggested by the reviewer.* |
| The conclusions are a nice summary of the paper, but need the paper to make sense of the equations. Thus, they could not really be read stand-alone. Not sure if this is a problem. | *Since the reviewer considers the conclusions to be a nice summary of the paper, I have made no changes here because the conclusions make sense. If I were to remove the equations and replace them with text, the conclusions would be as long as the paper.* |